# Evaluation of Extended-Spectrum Beta-Lactamase Resistance in Uropathogenic *Escherichia coli* Isolates from Urinary Tract Infection Patients in Al-Baha, Saudi Arabia

**DOI:** 10.3390/microorganisms11122820

**Published:** 2023-11-21

**Authors:** Samiyah Ahmad Abdullah Alghamdi, Shazia Shaheen Mir, Fayez Saad Alghamdi, Mohammad Abdul Majeed Mohammad Aref Al Banghali, Shaia Saleh Rajab Almalki

**Affiliations:** 1Laboratory Medicine Department, Faculty of Applied Medical Sciences, Al-Baha University, Al-Baha 65779, Saudi Arabia; 442019601@stu.bu.edu.sa; 2Medical Training and Education Department, King Fahad Hospital, Al-Baha 65732, Saudi Arabia; fayezsg@moh.gov.sa; 3Public Health Department, Faculty of Applied Medical Sciences, Al-Baha University, Al-Baha 65779, Saudi Arabia; mohammad.aref@bu.edu.sa

**Keywords:** extended-spectrum beta-lactamases, urinary tract infections, uropathogenic *E. coli* (UPEC) and antibiotic resistance

## Abstract

Urinary tract infections (UTIs) caused by extended-spectrum beta-lactamase (ESBL)-producing organisms are prevalent in both outpatient and inpatient settings, representing the most often encountered forms of infection. This research aimed to estimate the prevalence of ESBL-UTIs along with other uropathogens in the adult population and to assess the antibiotic activity against *Escherichia coli* extended-spectrum beta-lactamase (*E. coli* ESBL) isolates from patient samples in Al-Baha. A retrospective cross-sectional study included patients who presented to King Fahad Hospital in Al-Baha with clinical suspicion of UTI between 1 January 2019 and 30 September 2022. A total of 4406 urine samples with significant microbial growth were included in the scope of this investigation. A collective count of 1644 incidents of *Escherichia coli* (*E. coli*) was observed, wherein *E. coli* constituted 85% of the cases, while the remaining 15% comprised *E. coli* ESBL producers. The prevalence of *E. coli* ESBL was observed to be 64.7% in females and 35.3% in males, with a majority (67%) of the affected individuals being over the age of 50. The incidence of *E. coli* infections in the outpatient setting was found to be greater than that observed in the inpatient setting. *E. coli* ESBL were sensitive to colistin, tigecycline, amikacin, meropenem, imipenem, and nitrofurantoin by 100% and 93.3–100%, 95–99.6%, 95–99.06%, and 81–91%, respectively. On the other hand, the most resistant agents for *E. coli* ESBL were the group of cephalosporins, aztreonam, and ampicillin with 100% resistance, ciprofloxacin with 56–74% resistance, and cotrimoxazole with a 45–53% resistance level. ESBL-resistant *E. coli* strains are moderately prevalent in community- and hospital-acquired UTIs, especially in females and elderly patients (>50 years).

## 1. Introduction

Urinary tract infections (UTIs) are considered to be the second most common bacterial illness, behind respiratory infections. The conventional characterization of a urinary tract infection (UTI) is the presence of more than 100,000 colony-forming units (CFUs) per milliliter of urine, accompanied by the characteristic acute symptoms of suprapubic pain, frequency, or urgency. This description incorporates multiple manifestations of urinary tract infections, such as simple cystitis or uncomplicated cystitis, pyelonephritis, as well as the more serious situations of bacteremia and septicemia [1]. The average incidence of sepsis associated with urinary tract infections among hospitalized individuals is estimated to be around 42 percent [2]. The concern lies in the increasing antimicrobial resistance and its potential impact on patient safety. Both multidrug-resistant organisms and their effects on surveillance systems are areas of concern [3,4]. The emergence of antibiotic-resistant bacteria is widely recognized as a significant public health concern [5,6]. The issue of treating various illnesses is exacerbated by the increasing resistance observed in numerous countries, primarily due to the scarcity of new antibiotics. Uropathogenic *Escherichia coli* (UPEC) accounts for a significant proportion, ranging from 75% to 90%, of bacterial isolates responsible for urinary tract infections [7]. Among these infections, approximately 30–50% are acquired within healthcare settings (nosocomial), while approximately 80–90% are acquired within the community. The female population exhibited a UTI recurrence rate of 25% within six months. Infections caused by antimicrobial resistance (AMR) are responsible for the deaths of around 700,000 people throughout the world every year, and this number is anticipated to climb to 10 million by 2050 [8]. Geographical differences may have an impact on the prevalence rates of ESBL seen across various investigations. The estimation of incidences of antibiotic resistance or ESBLs might be influenced by the inclusion and exclusion criteria of isolates, leading to potential overestimation or underestimation [9]. The administration of conventional antibiotics as empirical treatment is often provided to the majority of patients upon receiving a diagnosis of urinary tract infections. UTIs caused by multidrug-resistant *Escherichia coli* able to produce extended-spectrum β-lactamases (ESBL-MDR-EC) are increasing worldwide. The ESBLs in multidrug-resistant organisms have been a significant obstacle for medical practitioners in the management of urinary tract infections; ineffective and delayed treatment can lead to serious clinical complications, such as sepsis, renal scarring, and prolonged hospitalization, compared to non-ESBL infections [7,10].

Extended-spectrum beta-lactamase-producing urinary tract infection (ESBL-UTI) organisms include heritable mobilized plasmids that encode for extended-spectrum beta-lactamases, resulting in an increased level of resistance, in part as a consequence of the use, abuse, and misuse of β-lactam antibiotics [11,12]. ESBL are enzymes that hydrolyze penicillin rings, first-, second-, and third-generation cephalosporins, and monobactams (aztreonam) [13]. Since their discovery, approximately 300 ESBL gene variants have been discovered in diverse *Enterobacteriaceae* family members and many more non-enteric organisms. ESBLs derived from different enzymes (SHV-, TEM-, and CTX-M-type) have been reported from the diagnostic *Enterobacteriaceae* strains, providing broader antimicrobial resistance comprising of fluoroquinolones, aminoglycosides, and β-lactams [14]. Antibiotic resistance genes in these bacteria often code for resistance to cotrimoxazole, quinolones, and aminoglycosides. These cases are categorized as MDRs (multidrug resistant). Hence, the choice of antibiotic regimen for ESBLs is often complicated and necessitates prompt identification of ESBL-producing isolated strains to provide effective therapy and disease control in healthcare settings [15].

The World Health Organization designated ESBL-producing *Enterobacteriaceae* (ESBL-EB) as the highest public health risk in its 2021 report. Nonetheless, there has been limited research into the specific geographical distribution of these bacteria, notably inside Saudi Arabia [16,17]. Therefore, it is important to have a comprehensive understanding of the prevalence of uropathogenic *E. coli* and its resistance to antibiotics in the local population. This knowledge is essential for timely diagnosis and the implementation of suitable treatment strategies to reduce the potential adverse consequences [18,19]. Hence, the primary objective of this research was to ascertain the frequency at which UTI-causing bacteria are present, as well as to assess the susceptibility of ESBL-producing *E. coli* strains causing UTI, specifically among patients at King Fahad Hospital. The findings of this research will be used to develop a program intended to address the incidence of bacterial resistance and establish regulations for the use of antibiotics.

## 2. Materials and Methods

### 2.1. Sample Size

A retrospective cross-sectional study was conducted using archived medical records and test outcomes of urine samples received by the microbiology laboratory for culture and sensitivity analysis at King Fahad Hospital in the Al-Baha region of Saudi Arabia. The data were gathered by the authors throughout the period spanning from January 2019 to September 2022.

### 2.2. Study Setting

This research was carried out at King Fahad Hospital in Al-Baha, Saudi Arabia, which has accreditation from CBAHI, the Central Board for Accreditation of Healthcare Facilities in Saudi Arabia. The hospital in question receives patient referrals from neighboring clinics, private hospitals, and health institutions in the vicinity.

### 2.3. Study Design

A cross-sectional retrospective analysis was conducted to analyze all urine samples for urinary tract infections that were submitted to the microbiology laboratory for culture and sensitivity testing from January 2019 to September 2022. This study collected data on patients’ information, identification of bacteria, and profiles of antibiogram AST from the OASIS program’s Electronic Medical Record (EMR) system. A standardized data-gathering sheet in Excel format was used for this purpose. The research undertaken at KFH investigated the prevalence of urinary tract infections in individuals across all age groups and genders, including both newly diagnosed cases and recurrent infections. This study also included patients receiving treatment as both inpatients and outpatients. The research specifically eliminated situations where there was a combination of growth and negative growth culture, as well as any possible risk variables that may be related to urinary tract infections.

### 2.4. Inclusion and Exclusion Criteria

All individuals, regardless of age or gender, who tested positive for urinary tract infections (UTIs) in both new cases and recurrent infections and in both inpatient and outpatient settings were included in this study conducted at KFH. All cultures exhibiting mixed growth and negative growth were excluded from this study.

### 2.5. Data Collection

The data variables were predetermined, and the data collection sheet was prepared and standardized to gather patient data from the OASIS program’s Electronic Medical Records (EMR) and laboratory results (Figure 1). The factors considered for data collection were age, sex, sample collection date, bacterial isolates, and antibiotic sensitivity pattern. The preservation of data integrity and confidentiality was ensured. The information that was gathered was only used for this research.

### 2.6. Urine Analysis

The collection and processing of samples were carried out in compliance with internal policy procedures (IPPs). The urine samples were incubated at a temperature of 37 °C for 24–48 h to facilitate bacterial culture. The aforementioned task was achieved by using a loop with a standardized volume of 1 L on plates that were supplemented with blood agar, MacConkey agar, and cystine–lactose–electrolyte-deficient (CLED) medium. Following that, the utilization of the Gram staining method was applied to discern between Gram-positive cocci and Gram-negative rods. The VITEK system was used to conduct antibiotic susceptibility and confirmation assays. The data were subjected to analysis and interpretation following the principles set out by the Clinical Laboratory Standards Institute (CLSI).

### 2.7. Statistical Analysis of Data

The database was created using MS Excel, while the data analysis was conducted using version 20 of the Statistical Package for the Social Sciences (SPSS Inc., Chicago, IL, USA). In order to describe the data, a descriptive analysis was performed. The use of measures of central tendency, such as the median and standard deviation, was employed for continuous data, and measures of dispersion, such as frequency and percentage, were utilized for categorical variables. The results of this study are visually represented via the use of graphs and tables. The findings of a retrospective analysis revealed the frequency of ESBL-producing *Escherichia coli* (*E. coli*) and their patterns of antibiotic susceptibility throughout the period from January 2019 to September 2022.

## 3. Results

### 3.1. Prevalence of Uropathogenic Bacteria

In this investigation, it was shown that *E. coli* emerged as the prevailing pathogen among a total of 4406 positive isolates, which included 1353 instances of recurring infections. A total of 1644 instances of *E. coli* were discovered, with *E. coli* accounting for 85% of the cases and *E. coli* ESBL producers accounting for the remaining 15% of the *E. coli* cases. The study participants were stratified into four distinct age groups, and it was observed that those aged 50 years and above had a greater incidence of urinary tract infections (UTIs) compared to those in younger age groups. The data were evenly distributed across different genders, age groups, patient settings, cases, and species of bacteria and fungus. This distribution is shown in Table 1 via tables and bar charts. The prevalence of uropathogenic bacteria, namely *E. coli*, *Klebsiella*, and *Candida* spp., was found to be greatest in both genders, with percentages of 31.59%, 16.82%, and 13.28%, respectively. The prevalence of ESBL-producing bacteria, namely *E. coli* and *K. pneumonia*, was found to be 5.72% and 1.23%, respectively, as shown in Table 1. Additionally, the presence of MRSA was observed to be just 0.89%. In the examined population, it is seen that the proportion of females (66.5%) surpasses that of men (33.5%). Furthermore, both genders exhibit a greater percentage of individuals aged over 50 compared to other age groups.

In the context of comparing the prevalence of uropathogens in inpatient and outpatient settings, it was shown that *Escherichia coli* exhibited the highest frequency among those receiving outpatient care. In contrast, it has been shown that *Klebsiella* spp. exhibits a greater incidence within the context of inpatient settings, with *Candida* spp. showing a similar pattern.

Table 2 shows the different percentages between both genders, ages, and types of cases, which indicates the instances of infection with *E. coli* ESBL in females is (64.7%) more than in males (35.3%). However, both genders >50 age group showed a similar rate of infections, while the new cases and recurrent infections were more common in females.

Figure 2 shows that *E. coli* ESBL percentage was more frequent with outpatient (143) than inpatient (109), which may indicate a community-acquired infection.

### 3.2. Antibacterial Susceptibility Patterns

The antibiotics that demonstrated the highest sensitivity against *E. coli* strains carrying extended-spectrum beta-lactamase (ESBL) during the period from January 2019 to September 2022 were colistin (100%), tigecycline (100%), amikacin (93.3–100%), meropenem (95–99.6%), imipenem (95–99.06%), nitrofurantoin (81–91%), levofloxacin (79–94%), augmentin (44–87%), norfloxacin (74–86%), and tazocin (60–66%). In contrast, the group of cephalosporins, aztreonam, and ampicillin exhibited complete resistance to *E. coli* with ESBL, whereas ciprofloxacin showed resistance rates ranging from 56% to 74%, and cotrimoxazole exhibited resistance rates ranging from 45% to 53% (Table 3, Table 4 and Table 5).

## 4. Discussion

### 4.1. Prevalence of Uropathogenic Bacteria

In this study, the most prevalent uropathogens were *E. coli*, *Klebsiella*, *Enterococcus*, and *Candida*, which were 31.59%, 16.82%, 8.78%, and 13.28%, respectively. On the other side, the percentages of *Staphylococcus*, *Streptococcus,* and *MRSA* infection were 5.29%, 5.11%, and 0.84%, respectively. Our findings are in concordance with previous research, which also found that UTI infections by Gram-negative bacteria outnumber Gram-positive bacteria [20,21,22,23,24]. *Enterobacteriaceae*-produced ESBLs have the potential to induce severe infections in individuals, afterward leading to the development of more intricate pathological conditions. The correlation between the existence of pathogenic bacteria that generate extended-spectrum beta-lactamase (ESBL) in an infection and adverse clinical consequences has been established. The aforementioned results encompass delayed clinical and microbiological responses, extended durations of hospitalization, elevated healthcare expenditures, and heightened rates of morbidity. Currently, the growing number of *Enterobacteriaceae* that make ESBLs is a global health concern because antibiotic resistance is high and treatment options are limited. This study identified 1644 cases of *E. coli*, of which *E. coli* accounted for 85% and *E. coli* ESBL producers for 15%. Other studies, such as Al Khawaja et al., 2019, found a higher percentage of *E. coli* ESBL producers, 27.39% of all *E. coli* cases. Two studies in Riyadh have yielded similar results, with *E. coli* ESBL producers comprising one-third of the total *E. coli* population [20,25]. However, our detection of an ESBL-producing *E. coli* strain was still lower than that found in other studies. This higher percentage of UTIs caused by *E. coli* ESBL-producing strains in those studies could be attributed to the heterogeneity of the studied individuals, as many risk factors for *E. coli* ESBL infection differ from patient to patient [25,26].

### 4.2. Socio-Demographic Risk Factors

The findings of this study revealed that females exhibited a greater prevalence of infection with *E. coli* ESBL producers across various age cohorts. In contrast, it was revealed that individuals of both genders demonstrated a greater prevalence of infection within age cohorts surpassing 50 years. This phenomenon may be attributed to the age-related decline in immune system function and heightened vulnerability to infections.

These findings align with multiple studies that have demonstrated a higher likelihood of uropathogenic bacterial colonization in women compared to men. This disparity can be attributed to various factors, such as women having shorter urethras and urethras located in closer proximity to the rectum. The latter is significant as uropathogenic *Escherichia coli* (UPEC) is primarily derived from fecal flora, rendering women more vulnerable to urethral infections in general and specifically to extended-spectrum beta-lactamase (ESBL)-producing *E. coli* [24,27,28,29]. As a consequence of hormonal, mechanical, and physiological alterations occurring during pregnancy, alongside fluctuations in estrogen levels and vaginal pH due to the aging process, it can be concluded that the musculature of the bladder and pelvic floor experiences a decline in strength as individuals grow older. This weakening often leads to urinary retention or incontinence, which in turn increases the likelihood of urinary tract infections (UTIs) [27,30,31,32].

### 4.3. Antibacterial Susceptibility Patterns

In the context of extended-spectrum beta-lactamase urinary tract infections (ESBL-UTI), the careful selection of antibiotics is critical. It is very important to use prudent discretion while prescribing them while considering the regional prevalence of bacterial species and susceptibility data. Given the limited availability of therapeutic interventions, these infections present a formidable obstacle to public health in both community and hospital environments, as they have the potential to cause substantial mortality and morbidity, particularly among individuals with compromised immune systems and the elderly. The substantial economic consequences of ESBL-MDR-EC infections have been ascribed to the elevated expenses associated with antibiotics and the extensive utilization of healthcare resources [33]. Therefore, it is important to consistently assess the antimicrobial susceptibility profile and engage in surveillance of resistance at many levels, including local, national, and global spheres. There exists an urgent necessity to assess the efficacy of prevailing therapeutic approaches, such as the use of cefuroxime, quinolones, cotrimoxazole, and augmentin, in light of the significant degree of resistance observed in the management of urinary tract infections [8].

The present investigation has shown that strains of *E. coli* that produce extended-spectrum beta-lactamases (ESBLs) exhibit a significant level of resistance to a wide range of antibiotics. It is comprehensible that *Escherichia coli* bacteria that produce extended-spectrum beta-lactamases (ESBLs) often exhibit resistance to other antimicrobial agents such as fluoroquinolones and aminoglycosides [34]. The antibiotics that showed great efficacy against *E. coli* ESBL were colistin, tigecycline, amikacin, meropenem, imipenem, nitrofurantoin, levofloxacin, augmentin, norfloxacin, and tazocin. In contrast, the cluster of cephalosporins, aztreonam, ampicillin, ciprofloxacin, and cotrimoxazole exhibited the highest levels of antibiotic resistance against *E. coli* with ESBL. This finding aligns with other previous studies that have shown that *E. coli* isolates containing ESBLs exhibit higher levels of resistance to TMP-SXT, ciprofloxacin, cotrimoxazole, gentamicin, augmentin, and third-generation cephalosporins compared to *E. coli* isolates that do not manufacture ESBLs. Nevertheless, this resistance did not have an impact on the effectiveness of carbapenems [12,20,24,35]. Incorrect use, heavy abuse, or self-medication when treating UTIs may result in higher antibiotic-resistant strains [36]. Furthermore, urinary tract infections (UTIs) have been shown to result in elevated levels of morbidity and death, particularly in instances of bacteremia. The current patterns of antibiotic-resistant organisms have further complicated the management of these infections [37]. Our recommendation for first-line empirical therapy for UTIs includes carbapenems, amikacin, colistin, and nitrofurantoin, with the latter being less preferred due to its comparatively higher incidence of side effects [21,30]. It is not advisable to use any antibiotic as empirical treatment for lower urinary tract infections if the local resistance exceeds 20%. Similarly, for upper urinary tract infections, the threshold for local resistance should not exceed 10%. [38]. The antibiotic susceptibility profile of *E. coli* ESBL producers, in general, shows a modest trend, characterized by a significant rise in resistance to ciprofloxacin, ranging from 56% to 70% in recent years. This observation aligns with the findings of recent research conducted by van Driel in 2020 [39].

However, contrary to many studies, our results showed a decrease in cases of *E. coli* ESBL-producer infection [20,24,36,39,40]. According to the statistics presented, there is a steady annual decline in the incidence of *E. coli* ESBL producers. Several factors contribute to this phenomenon, including the Ministry of Health’s (MOH) policy that prohibits the purchase of antibiotics without a prescription from a physician at commercial pharmacies. Additionally, the annual program known as World Antibiotic Awareness Week plays a significant role in increasing patient awareness regarding the appropriate use of antibiotics. Furthermore, the effective surveillance of microbial resistance to antibiotics, as evidenced by the antibiogram conducted in microbiology laboratories, along with the implementation of the Antimicrobial Stewardship Program at KFH, further contributes to this outcome [37]. Moreover, it is plausible that the COVID-19 epidemic and the implementation of social distancing measures have indirectly contributed to the notable decline in isolated *E. coli* ESBL cases. These circumstances have hindered the ability of many individuals to seek medical attention at healthcare facilities. There is a lack of research conducted in the Al-Baha area that provides evidence either supporting or refuting the impact of these choices and the adoption of Antimicrobial Stewardship Programs (ASPs) on the reduction in cases infected with antibiotic-resistant strains of bacteria in both community and healthcare settings [41].

Evaluating regional or institutional antimicrobial resistance frequencies is crucial for devising and updating the current recommendations for rapid and optimal treatment protocols that efficiently manage ESBL-producing uropathogens, particularly *E. coli*, that lead to uncomplicated urinary tract infections.

## 5. Conclusions

In this study, we conclude that the resistant strains of *E. coli* ESBL were moderately prevalent in the community- and hospital-acquired UTIs, particularly among female patients. Levofloxacin, colistin, tigecycline, amikacin, meropenem, imipenem, and nitrofurantoin were effective against *E. coli* ESBL. These findings provide additional evidence that it is necessary to utilize earlier and more precise methods for identifying ESBL-producing bacteria to appropriately prescribe effective antibiotics.

## 6. Strengths and Limitations of This Study

This study provides important information on the prevalence of different uropathogens and antibiotic resistance in ESBL *E. coli*. These data are crucial for identifying the microorganism and guiding empirical therapy for urinary tract infections, particularly in the Al-Baha region. The large sample size of more than 4000 patients experiencing UTI episodes greatly increased the statistical power of the studies. This study provides a comprehensive view of ESBL *E. coli* in the Al-Baha general population. Our claim is supported by the fact that King Fahad Hospital, which implements the Antibiotic Stewardship Program, is a referral hospital where most people in Al-Baha seek medical care. This study also contributes to the distinction of uropathogens distribution patterns in community-acquired and nosocomial urinary tract infections (UTIs). As a consequence, it is feasible to determine the frequency of bacteria and ESBL *E. coli* resistance patterns in both types of urinary tract infections.

Our main limitations are typical of those of retrospective studies using routinely collected electronic health record data. Missing data, residual confounders, and potential biases are issues. Therefore, certain numbers may be skewed and not entirely reflective of the exact patient demographics. The laboratory environment posed time restrictions that limited the feasibility of determining ESBL genotyping in *E. coli* strains. To better understand the resistance mechanism and its prevalence in the area, further research on genotyping of ESBL *E. coli* should be undertaken. This study is limited to the Al-Baha region; therefore, it does not sufficiently represent the wide range of cyclic, pathological, and cultural changes exhibited by uropathogenic bacteria, as well as their corresponding antimicrobial sensitivity profiles, across the entire country.

## Figures and Tables

**Figure 1 microorganisms-11-02820-f001:**
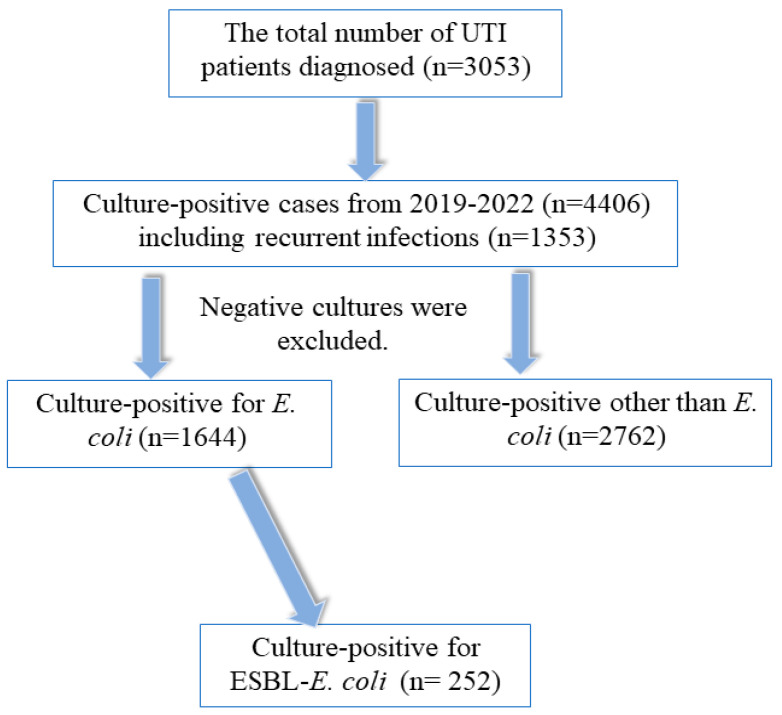
Flowchart for patient selection.

**Figure 2 microorganisms-11-02820-f002:**
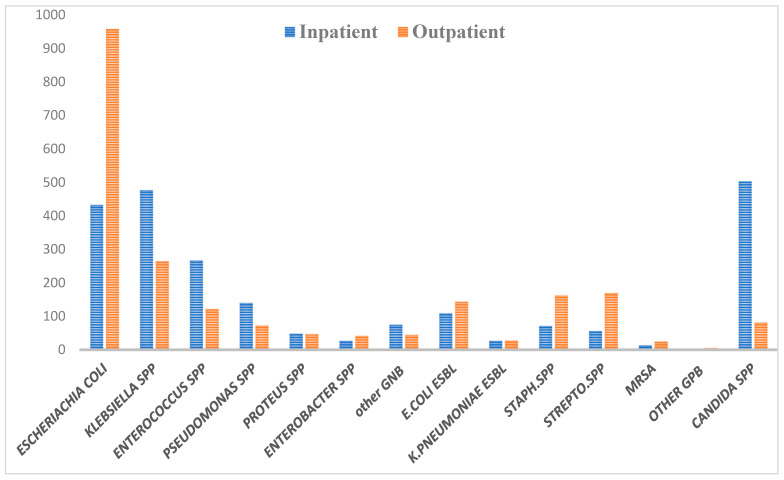
The frequencies of the uropathogens according to patient setting (in\outpatient).

**Table 1 microorganisms-11-02820-t001:** The distribution of age groups and gender with positive urine culture from January 2019 to September 2022.

Variable	Male N (%)	Female N (%)	Total N (%)
Age group	1022 (33.5)	2031 (66.5)	3053 (100%)
Median (in years)	48 (SD +\− 30.65)	48 (SD +\− 30.62)	48 (SD +\− 30.67)
<18	220 (21.53)	509 (25.10)	729 (23.87)
18–24	33 (3.23)	110 (5.42)	143 (4.70)
25–49	138 (13.50)	545 (26.80)	683 (22.37)
>50	631 (61.74)	867 (42.68)	1498 (49.06)
Culture positives (including recurrent cases)	1524 (34.59)	2882 (65.41)	4406 (100)
**GNB**			
*Escherichia coli*	339 (22.24)	1053 (36.54)	1392 (31.59)
*Klebsiella* spp.	290 (19.03)	451 (15.65)	741 (16.82)
*Enterobacter* spp.	32 (2.10)	37 (1.28)	69 (1.57)
*Pseudomonas* spp.	109 (7.15)	103 (3.57)	212 (4.81)
*Proteus* spp.	48 (3.15)	46 (1.60)	94 (2.13)
Other GNB	57 (3.7)	62 (2.2)	119 (2.7)
ESBL producer			
*E. coli ESBL*	89 (5.84)	163 (5.66)	252 (5.72)
*K. pneumoniae ESBL*	20 (1.31)	34 (1.18)	54 (1.23)
**GPB**			
*Staph.* spp.	105 (6.89)	128 (4.44)	233 (5.29)
*Strep.* spp.	31 (2.03)	194 (6.73)	225 (5.11)
*MRSA*	12 (0.79)	25 (0.87)	37 (0.84)
*Enterococcus* spp.	170 (11.15)	217 (7.53)	387 (8.78)
Other GPB	1 (0.1)	5 (0.2)	6 (0.1)
**Fungi**			
*Candida* spp.	221 (14.50)	364 (12.63)	585 (13.28)

GNB, Gram-negative bacteria; GPB, Gram-positive bacteria; ESBL, extended-spectrum beta-lactamase.

**Table 2 microorganisms-11-02820-t002:** The different percentages of isolated *E. coli* ESBL between age, gender, and type.

Age Group	Male N (%)	Female N (%)	Total N (%)
<18	14 (15.7)	31 (19.0)	45 (17.8)
18–24	5 (5.6)	4 (2.5)	9 (3.7)
25–49	5 (5.6)	24 (14.7)	29 (11.5)
>50	65 (73.1)	104 (63.8)	169 (67)
Total cases of *E. coli* ESBL	89 (35.3)	163 (64.7)	252 (100)
New cases	54 (60.7)	123 (75.5)	177 (70.2)
Recurrent cases	35 (39.3)	40 (24.5)	75 (29.8)

ESBL: extended-spectrum beta-lactamase.

**Table 3 microorganisms-11-02820-t003:** The pattern of *AST* of *E. coli* ESBL with β-lactam (cephalosporins and monobactam) R.N., resistance number; S.N., sensitive number.

	Variables	CEP	CTR	CTX	CAZ	CXM	CFPM	ATM
**2019**	R.N. (%)	120	120	120	120	120	120	120
		(100%)	(100%)	(100%)	(100%)	(100%)	(100%)	(100%)
	S.N. (%)	0	0	0	0	0	0	0
		(0%)	(0%)	(0%)	(0%)	(0%)	(0%)	(0%)
	Total	120	120	120	120	120	120	120
**2020**	R.N. (%)	58	58	58	58	57	58	58
		(100%)	(100%)	(100%)	(100%)	(98%)	(100%)	(100%)
	S.N. (%)	0	0	0	0	1	0	0
		(0%)	(0%)	(0%)	(0%)	(2%)	(0%)	(0%)
	Total	58	58	58	58	58	58	58
**2021**	R.N. (%)	38	38	38	38	38	38	38
		(100%)	(100%)	(100%)	(100%)	(100%)	(100%)	(100%)
	S.N. (%)	0	0	0	0	0	0	0
		(0%)	(0%)	(0%)	(0%)	(0%)	(0%)	(0%)
	Total	38	38	38	38	38	38	38
**2022**	R.N. (%)	36	36	36	36	36	36	36
		(100%)	(100%)	(100%)	(100%)	(100%)	(100%)	(100%)
	S.N. (%)	0	0	0	0	0	0	0
		(0%)	(0%)	(0%)	(0%)	(0%)	(0%)	(0%)
	Total	36	36	36	36	36	36	36

Abbreviations: R, resistance; S, sensitive; CEP, cephalothin; CTR, ceftriaxone; CTX, cefotaxime; CAZ, ceftazidime; CXM, cefuroxime; CFPM, cefepime; ATM, aztreonam.

**Table 4 microorganisms-11-02820-t004:** The pattern of AST of *E. coli* ESBL with β-lactam (penicillin, carbapenem) and aminoglycosides.

	Variable	AMP	TAZ	AUG	MEM	IMI	AMK	GEN
**2019**	R.N. (%)	120	45	49	1	2	8	44
		(100%)	(37.5%)	(40.8%)	(0.94%)	(1.7%)	(6.6%)	(36.6)
	S.N. (%)	0	75	71	119	118	112	76
		(0%)	(62.5%)	(59.2%)	(99.06%)	(98.3%)	(93.3%)	(63.4%)
	Total	120	120	120	120	120	120	120
**2020**	R.N. (%)	58	23	35	1	1	0	6
		(100%)	(40%)	(60%)	(2%)	(2%)		(11%)
	S.N. (%)	0	35	23	57	57	58	52
		(0%)	(60%)	(40%)	(98%)	(98%)	(100%)	(89%)
	Total	58	58	58	58	58	58	58
**2021**	R.N. (%)	38	13	5	2	2	0	3
		(100%)	(34%)	(13%)	(5%)	(5%)		(7%)
	S.N. (%)	0	25	33	36	36	38	35
		(0%)	(66%)	(87%)	(95%)	(95%)	(100%)	(93%)
	Total	38	38	38	38	38	38	38
**2022**	R.N. (%)	36	14	20	1	1	0	4
		(100%)	(39%)	(56%)	(3%)	(3%)		(11%)
	S.N. (%)	0	22	16	35	35	36	32
		(0%)	(61%)	(44%)	(97%)	(97%)	(100%)	(89%)
	Total	36	36	36	36	36	36	36

Abbreviations: R, resistance; S, sensitive; AMP, ampicillin; TAZ, tazocin; AUG, augmentin; MEM, meropenem; IMI, Imipenem; AML, amikacin; GEN, gentamicin.

**Table 5 microorganisms-11-02820-t005:** The pattern of AST of *E. coli* ESBL (Fluoroquinolone, Polymyxin, Glycylglycine, Sulfonamides, and Nitrofuran).

	Variable	LVX	NOR	CPFX	CST	TGC	CMX	NFN
**2019**	R.N. (%)	18	30	68	0	0	55	16
		(15%)	(25%)	(56.6)			(45.8%)	(13%)
	S.N. (%)	102	90	52	120	120	65	104
		(85%)	(75%)	(43.4%)	(100%)	(100%)	(54.2%)	(87%)
	Total	120	120	120	120	120	120	120
**2020**	R.N. (%)	3	8	36	0	0	31	5
		(5.2%)	(13%)	(62%)			(53.5%)	(9%)
	S.N. (%)	55	50	22	58	58	27	53
		(94.8%)	(86%)	(38%)	(100%)	(100%)	(46.5%)	(91%)
	Total	58	58	58	58	58	58	58
**2021**	R.N. (%)	8	10	28	0	0	20	7
		(21%)	(26%)	(74%)			(53%)	(18%)
	S.N. (%)	30	28	10	38	38	18	31
		(79%)	(74%)	(26%)	(100%)	(100%)	(47%)	(82%)
	Total	38	38	38	38	38	38	38
**2022**	R.N. (%)	5	6	25	0	0	19	7
		(14%)	(17%)	(70%)			(53%)	(19%)
	S.N. (%)	31	30	11	36	36	17	29
		(86%)	(83%)	(30%)	(100%)	(100%)	(47%)	(81%)
	Total	36	36	36	36	36	36	36

Abbreviations: R, resistance; S, sensitive; LVX, levofloxacin; NOR, norfloxacin; CPFX, ciproflaxin; CST, colistin; TGC, tigecycline; CMX, cotrimoxazole; NFN, nitrofurantoin.

## Data Availability

The data presented in this study are available on request from the corresponding author. This research contained no personally identifiable information. Data from routine monitoring that had been anonymized were used in this secondary analysis.

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
