# Peer review of "Evaluation of Extended-Spectrum Beta-Lactamase Resistance in Uropathogenic Escherichia coli Isolates from Urinary Tract Infection Patients in Al-Baha, Saudi Arabia"

_microorganisms, 2023, doi:10.3390/microorganisms11122820_

Round 1
Reviewer 1 Report
Comments and Suggestions for Authors The article under review is devoted to the study of uropathogenic and antibiotic-resistant E. coli strains. Overall, the article is well written, a study that is certainly important for practical medicine. The methods are adequate to the task, the conclusions correspond to the results obtained.In principle, the article can be published, but I have a few comments: 1. In my opinion, the title of the article is unfortunate; in addition, I would recommend avoiding abbreviations in the title. 2. I would not divide the abstract into sections. 3. In materials and methods: inclusion and exclusion criteria can be merged.
Author Response
Thank you very much for your comments and giving us a chance to revise the manuscript with Manuscript ID: microorganisms-2616816 entitled “An Evaluation of Uropathogenic Escherichia coli with ESBL Resistance Strains Among UTI Patients in Al Baha, Saudi Arabia”. Your comments have indeed increased the quality of this research article. We have revised the manuscript according to the comments provided by the reviewer’s and have highlighted the changes in red throughout the manuscript. We have also added some new literature with specific citations in response to the comments provided by the reviewer’s.
Reviewer #1:
The article under review is devoted to the study of uropathogenic and antibiotic-resistant E. coli strains. Overall, the article is well written, a study that is certainly important for practical medicine. The methods are adequate to the task, the conclusions correspond to the results obtained.
Comment 1: In my opinion, the title of the article is unfortunate; in addition, I would recommend avoiding abbreviations in the title.
Response: Thank you very much for your comment. The authors have changed the title and removed the abbreviations from the title. The new title is as follows: “Evaluation of Extended-Spectrum Beta-Lactamase Resistance in Uropathogenic Escherichia coli Isolates from Urinary Tract Infection Patients in Al Baha, Saudi Arabia”
Comment 2: I would not divide the abstract into sections.
Response: Thank you for your comment. In the abstract, the separate sections have been removed and are now represented as one paragraph.
Comment 3: In materials and methods, inclusion and exclusion criteria can be merged.
Response: In materials and methods: inclusion and exclusion criteria have been merged and presented in one single sub-heading, “Inclusion and Exclusion Criteria.”
Reviewer 2 Report
Comments and Suggestions for Authors
Urinary tract infections represent an intensely debated subject worldwide.
Determining local prevalence and resistance patterns may be a useful tool in medical practice.
The present study is well-designed overall but there are some aspects that might be improved.
1. It may be useful to insert a chart for patient selection in the material and methods section.
2. The discussion section should be divided into small chapters to be more easy to follow.
3. Please add strengths and limitations to the present study.
4. The conclusion section is too long. Is should be more concise and to highlight the main idea.
Comments on the Quality of English Language
No major English issues identified.
Author Response
Thank you very much for your comments and giving us a chance to revise the manuscript with Manuscript ID: microorganisms-2616816 entitled “An Evaluation of Uropathogenic Escherichia coli with ESBL Resistance Strains Among UTI Patients in Al Baha, Saudi Arabia”. Your comments have indeed increased the quality of this research article. We have revised the manuscript according to the comments provided by the reviewer’s and have highlighted the changes in red throughout the manuscript. We have also added some new literature with specific citations in response to the comments provided by the reviewer’s.
Reviewer #2:
Determining local prevalence and resistance patterns may be a useful tool in medical practice. The present study is well-designed overall, but there are some aspects that might be improved.
Comment 1: It may be useful to insert a chart for patient selection in the material and methods section.
Response: Thank you for your comment. A flowchart for patient selection has been included in the material and methods and is represented in Figure 1.
Comment 2: The discussion section should be divided into small chapters to be easier to follow.
Response: Thank you for your comment. The discussion section has been divided into small chapters.
Comment 3: Please add strengths and limitations to the present study.
Response: Thank you for your comment. After the conclusion, a separate section discussing the current study's strengths and limitations has been added.
Comment 4: The conclusion section is too long. It should be more concise and highlight the main idea.
Response: The conclusion section has been shortened to highlight the main idea while being brief.